# Self Fourier shell correlation: properties and application to cryo-ET

Eric J. Verbeke [1✉], Marc Aurèle Gilles[1], Tamir Bendory[2] & Amit Singer [1,3]

The Fourier shell correlation (FSC) is a measure of the similarity between two signals computed over corresponding shells in the frequency domain and has broad applications in microscopy. In structural biology, the FSC is ubiquitous in methods for validation, resolution determination, and signal enhancement. Computing the FSC usually requires two independent measurements of the same underlying signal, which can be limiting for some applications. Here, we analyze and extend on an approach to estimate the FSC from a single measurement. In particular, we derive the necessary conditions required to estimate the FSC from downsampled versions of a single noisy measurement. These conditions reveal additional corrections which we implement to increase the applicability of the method. We then illustrate two applications of our approach, first as an estimate of the global resolution from a single 3-D structure and second as a data-driven method for denoising tomographic reconstructions in electron cryo-tomography. These results provide general guidelines for computing the FSC from a single measurement and suggest new applications of the FSC in microscopy.

[1] Program in Applied and Computational Mathematics, Princeton University, Princeton, NJ, USA. [2] School of Electrical Engineering, Tel Aviv University, Tel Aviv, Israel. [3] Department of Mathematics, Princeton University, Princeton, NJ, USA. ✉email: ev9102@princeton.edu

The Fourier shell correlation (FSC) is defined as the normalized cross-correlation of corresponding shells between two signals in the frequency domain[1,2]. In single particle electron cryo-microscopy (cryo-EM), the FSC has become the universal resolution metric and is used to assess the quality of a 3-D reconstruction[3,4]. Additional major contributions of the FSC in cryo-EM include setting hyperparameters of iterative algorithms, as in 3-D refinement of structures[5], and estimation of the spectral signal-to-noise ratio (SSNR)[6,7].

A core requirement of the FSC is the availability of two or more independent noisy measurements. In single particle cryo-EM, this is often achieved by splitting the data into random half sets[3]. However, for other forms of microscopy or data processing procedures, it is not always possible to apply the same strategy. To bypass the need for multiple measurements, a novel approach was recently proposed to estimate an FSC-like quantity from a single measurement, which we refer to as the self FSC (SFSC)[8]. This approach was initially used for image restoration in fluorescence microscopy[8] and has also been applied to estimating resolution in scanning electron microscopy[9]. The SFSC is implemented by first decimating an image in real space to produce downsampled images whose correlation with each other is then computed in Fourier space. While the interpretability of the original FSC has been discussed in refs. [10,11], the validity of the SFSC as a proxy for the FSC is more difficult to interpret, since the two downsampled signals are not independent.

In this work, we analyze the SFSC and give sufficient conditions on the statistics of both the signal and the noise under which the estimator is consistent with the standard FSC. Notably, we show that the assumptions required for the SFSC are more restrictive than the standard FSC and that use of the SFSC outside the defined conditions can give estimates that deviate significantly from the FSC. The conditions are easy to check and give practical guidelines to the applicability of the SFSC.

To demonstrate the validity of the SFSC, we provide two applications in the context of cryo-EM: first as a measure of the global resolution from a single map, and second as a data-driven method for denoising in electron cryo-tomography (cryo-ET). In the first application, we show that the resolution predicted by the SFSC from one half-map agrees with the standard FSC computed from two half maps, provided the conditions on the data described in this work are met. We then use the SFSC to denoise a reconstructed tomogram from cryo-ET data by applying a Wiener filter. Our approach provides significantly increased contrast and visibility compared to conventional low-pass filtering. The code used to generate the results in this work is available at: github.com/EricVerbeke/self_fourier_shell_correlation.

## Results

Experimental evidence for the relationship between the correlation of noisy image measurements and the signal-to-noise ratio in electron microscopy date back to (at least) 1975[12]. With the advent of the FSC, this relation was developed further to describe the decay in data quality with respect to spatial frequency based on a relation to the SSNR[6]. The SSNR is a central quantity in computational microscopy and has specific use in cryo-EM for denoising by Wiener filtering[13], and post-processing (e.g., 3-D structure sharpening)[3].

In this work, we consider the following simple model for estimating the FSC and therefore also the SSNR: we observe a single noisy measurement $y$ of an underlying signal $x$:

$$y = x + \epsilon. \quad (1)$$

We discuss the effect of including the contrast transfer function (CTF) in the model in Supplementary Note 1, and show the effect on the FSC in Supplementary Fig. S1. For the model in Eq. (1), we assume that the ground truth signal is drawn from a mean-zero Gaussian distribution $x \sim \mathcal{N}(0, \Lambda)$ with additive Gaussian colored noise $\epsilon \sim \mathcal{N}(0, \Sigma)$, where $\Lambda$ and $\Sigma$ are the covariance matrices of the signal and noise respectively. We further assume that all entries of $\hat{x} := \mathcal{F}\{x\}$ and $\hat{\epsilon} := \mathcal{F}\{\epsilon\}$, the discrete Fourier transforms (DFT) of $x$ and $\epsilon$, are statistically independent of each other, and thus have diagonal covariance matrices in the Fourier domain. That is, we have that $\hat{x} \sim \mathcal{N}(0, D(\lambda^2))$ and $\hat{\epsilon} \sim \mathcal{N}(0, D(\sigma^2))$, where $D(v) \in \mathbb{C}^{d \times d}$ denotes a diagonal matrix with entries $v \in \mathbb{C}^d$, and $\lambda$ and $\sigma$ are vectors that are constant along frequency shells. The real space covariance matrices are therefore $\Lambda = F^* D(\lambda^2) F$ and $\Sigma = F^* D(\sigma^2) F$, where $F$ is the normalized DFT matrix and $^*$ denotes the conjugate transpose. While the independence assumption on the signal in the Fourier domain may seem restrictive, it is typical in cryo-EM 3-D reconstruction—corresponding to a weighted $L^2$ regularized problem in the maximum a posteriori formulation[5]; furthermore, it is justified in the "infinitely large" protein limit under the Wilson statistics model[14,15]. The SSNR at spatial frequencies of radius $r$ is then defined as:

$$\mathrm{SSNR}(r) = \frac{\lambda^2(r)}{\sigma^2(r)}. \quad (2)$$

Typically, neither $\lambda^2$ or $\sigma^2$ are known a priori and thus must be estimated from data. In practice, the SSNR can be estimated if two independent and noisy measurements, $y_1$ and $y_2$, of the same signal $x$ are available by computing their FSC. The FSC is defined as:

$$\mathrm{FSC}(r) = \frac{\sum\limits_{k \in \mathcal{S}_r} \mathrm{Re}\left(\overline{\hat{y}_1[k]}\hat{y}_2[k]\right)}{\sqrt{\sum\limits_{k \in \mathcal{S}_r} |\hat{y}_1[k]|^2 \sum\limits_{k \in \mathcal{S}_r} |\hat{y}_2[k]|^2}} = \frac{\langle \hat{y}_1, \hat{y}_2 \rangle_r}{\| \hat{y}_1 \|_r \| \hat{y}_2 \|_r}, \quad (3)$$

where $\hat{y}_1$ and $\hat{y}_2$ are the DFT of $y_1$ and $y_2$, the overline denotes the complex conjugate, and $\langle \cdot, \cdot \rangle_r$ denotes the standard inner product on $\mathbb{C}^d$ restricted to the shell $\mathcal{S}_r$ with $\| \cdot \|_r$ being the associated norm. We use brackets to denote indexing of a discrete function and $k$ for the multi-index on a Fourier grid. The link between the FSC and SSNR is made by considering a related deterministic quantity, denoted EFSC:

$$\mathrm{EFSC}(r) := \frac{\mathbb{E}\left[\langle \hat{y}_1, \hat{y}_2 \rangle_r\right]}{\sqrt{\mathbb{E}[\| \hat{y}_1 \|_r^2]\mathbb{E}[\| \hat{y}_2 \|_r^2]}} = \frac{\lambda^2(r)}{\lambda^2(r) + \sigma^2(r)}, \quad (4)$$

where $\mathbb{E}$ is the expectation. We note that while $\mathbb{E}[\mathrm{FSC}(r)] \neq \mathrm{EFSC}(r)$, the estimated quantity has proven to be a useful proxy for the SSNR. From the EFSC, we see that the SSNR can be estimated as:

$$\mathrm{SSNR}(r) = \frac{\mathrm{EFSC}(r)}{1 - \mathrm{EFSC}(r)}. \quad (5)$$

It is a common practice to replace the EFSC by the empirical FSC Eq. (3) computed from two signals to estimate the SSNR.

**Fourier shell correlation from a single measurement.** The computation of the standard FSC requires two independent measurements of a signal. In this work, the goal is to estimate the FSC and SSNR from a single measurement. A solution proposed in[8] is to compute the FSC from downsampled versions of the same measurement. This approach is originally implemented by first taking a noisy, real-space measurement and decimating into a checkerboard-like pattern to form half-sized approximations of the original measurement, as shown in Supplementary Fig. S2. The FSC between pairs of downsampled signals can then be computed.

Here, we modify the downsampling procedure such that the real-space measurements are split into even and odd terms along one spatial dimension at a time, thus providing two downsampled versions for each dimension. The FSC is then computed between each downsampled measurement pair for each dimension and the reported FSC is taken to be the average, as shown in Fig. 1. There are two main advantages of this approach compared to the checkerboard-like splitting pattern. First, our scheme preserves the Nyquist frequency for each axis except the one split into even and odd terms. Second, we show in Supplementary Note 2 that splitting in a checkerboard-like pattern scales the variance of the noise in the SFSC by $2^{\text{dim}}$ where dim is the number of axes split into even and odd terms. For the 1-D case, a measured signal is decimated by simply splitting into even and odd terms.

While this process is always easily computable, it is not clear that the estimate is meaningful. Indeed, the basis of the connection between the FSC and SSNR is the statistical independence of two measurements. However, in the SFSC case, the two measurements are simply downsampled versions of the same noisy measurement which are correlated in any practical scenario. Despite the apparent correlation, we show that under conditions on the statistics of both the signal and the noise, the SFSC may still be used to estimate the SSNR from the

downsampled measurement, which can be used to infer the SSNR of the original measurement.

**Conditions for accurate estimation of the FSC from the SFSC.** We present our main analysis for the SFSC here using the one-dimensional case for simplicity, although we show in Supplementary Note 3 that it naturally extends to higher dimensions. Following the model in Eq. (1), let $y$ be a discrete 1-D measurement of length $N$, where we assume $N$ is even. The measurment $y$ is then downsampled by splitting it into even index terms $y_e[n] = y[2n]$ and odd index terms $y_o[n] = y[2n + 1]$ for $n \in \{0, \ldots, (N/2) - 1\}$. The DFT of the even and odd term measurements can be related to the DFT of the original measurement $y$ as follows (see Supplementary Note 3 for derivation):

$$\hat{y}_e[k] = (\hat{x}[k] + \hat{\epsilon}[k] + \hat{x}[k + N/2] + \hat{\epsilon}[k + N/2])/2, \quad (6)$$

$$\hat{y}_o[k] = (\hat{x}[k] + \hat{\epsilon}[k] - \hat{x}[k + N/2] - \hat{\epsilon}[k + N/2])/(2\omega_N^k), \quad (7)$$

where $\hat{y}_e$ and $\hat{y}_o$ are the DFTs of $y_e$ and $y_o$, and $\omega_N = \exp(-2\pi i/N)$. We note that if the higher frequency terms are small (i.e., there is a rapid decay in the power spectrum), then $\hat{y}_e[k]$ and $\hat{y}_o[k]$ are approximately equal after a phase shift of $\hat{y}_o[k]$ by $\omega_N^k$. Thus, as noted in[8], when computing the SFSC between downsampled pairs, a phase shift correction must be included. That is:

$$\text{SFSC}(r) = \frac{\left\langle \hat{y}_e, \hat{y}_o e^{-2\pi i \langle a, k/N \rangle} \right\rangle_r}{\| \hat{y}_e \|_r \| \hat{y}_o e^{-2\pi i \langle a, k/N \rangle} \|_r}, \quad (8)$$

where $a$ denotes the translation. We discuss the origin and effect of this translation further in Supplementary Note 4.

Our goal is to show that the SFSC is approximately equal to the FSC such that it can also provide an estimate of the SSNR as in Eq. (5). Following the same arguments as stated for the EFSC in Eq. (4), we have that:

$$\text{ESFSC}_{1\text{-D}}[k] := \frac{\mathbb{E}\left[\left\langle \hat{y}_e, \hat{y}_o e^{-2\pi i \langle a, k/N \rangle} \right\rangle_r\right]}{\sqrt{\mathbb{E}[\| \hat{y}_e \|_r^2]\mathbb{E}[\| \hat{y}_o e^{-2\pi i \langle a, k/N \rangle} \|_r^2]}}. \quad (9)$$

Under the standard assumption that the signal and the noise are statistically independent, we then get:

$$\text{ESFSC}_{1\text{-D}}[k] = \frac{\lambda^2[k] - \lambda^2[k + N/2] + \sigma^2[k] - \sigma^2[k + N/2]}{\lambda^2[k] + \lambda^2[k + N/2] + \sigma^2[k] + \sigma^2[k + N/2]}. \quad (10)$$

Thus, in general, the estimates for the EFSC and ESFSC are not the same. However, if we consider the following two assumptions:

Assumption 1: The Gaussian noise distribution is white, namely $\sigma^2[k] = \sigma^2[0] \; \forall \, k$,

Assumption 2: The power spectrum of the signal decays such that $\lambda^2[k] \gg \lambda^2[k + N/2]$,
     then, we have that:

$$\text{ESFSC}_{1\text{-D}}[k] \approx \frac{\lambda^2[k]}{\lambda^2[k] + 2\sigma^2[k]}. \quad (11)$$

That is, the ESFSC approximates the EFSC with an additional doubling on the variance of the noise. Given the above assumptions are met, the ESFSC can be related to the EFSC by:

$$\text{EFSC}(r) = \frac{2 \, \text{ESFSC}(r)}{1 + \text{ESFSC}(r)}. \quad (12)$$

We illustrate the importance of the assumptions on the signal and noise in Fig. 2, and the effect of phase shift and variance correction in Fig. 3a, b using a synthetic 2-D image as an

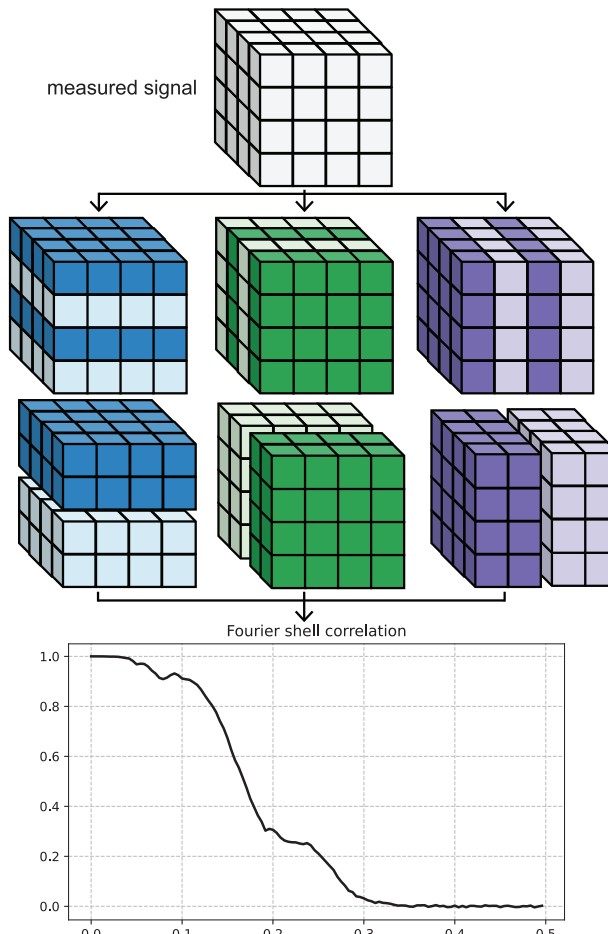

**Fig. 1 Illustration of the Fourier shell correlation computed from a downsampled signal (i.e., the SFSC).** The measured signal is split into even and odd voxels for each dimension and the SFSC is computed between the respective pairs. The reported FSC is taken to be the average of the three pairs.

example. The clean image originates from a projection of the 3-D structure of a human gamma-aminobutyric acid receptor (available as entry EMD-11657 in the electron microscopy data bank)[16]. Using a typical B-factor decay in cryo-EM[3], we generate each image as $x = \mathcal{F}^{-1}\{\hat{x}\exp(-B\parallel k\parallel^2/4VN)\}$, where $V$ is the voxel (or pixel) size and $B$ modulates the decay. Here the image size is $N \times N = 360 \times 360$ with a pixel size of 0.81 Å. Noisy measurements are then produced by adding Gaussian noise. To generate noise which decays with spatial frequency (i.e., such that Assumption 1 is broken), a B-factor decay may also be applied to the additive noise and is delineated as $B_{signal}$ and $B_{noise}$ when necessary.

Given both assumptions on the data are met, and that the correction for the scaling of the variance in Eq. (12) is applied, we show in Fig. 2a that the SFSC accurately estimates the FSC. If the noise is not white Gaussian, but instead decays with spatial frequency, the SFSC is unreliable and overestimates the FSC (Fig. 2b). However if the noise is white Gaussian, but the power spectrum of the signal does not have rapid decay, then the SFSC

equal to the SSNR. In fact, this is always possible and yields approximately equivalent results to the standard FSC. The noise level is typically computed as part of the cryo-EM reconstruction process and could be used for a more direct measure of the SSNR without the need of the FSC. We discuss this further in Supplementary Note 5 and demonstrate the simpler approach for estimating the SSNR in Supplementary Fig. S3.

**SFSC for measurements with slow decaying spectrum.** Assumption 2 requires that there is rapid decay in the power spectrum of the underlying signal. If this is not the case, we can introduce an additional correction to the SFSC. We propose the following approach: upsample the measurement by zero-padding in Fourier space to increase the length of the measurement to $\tilde{N} = 2N$, then subtract off the noise level from the numerator. The effect of the zero-padding is to set the high frequency terms to zero, and thus their variance is also zero (i.e., $\lambda^2[k + \tilde{N}/2] = \sigma^2[k + \tilde{N}/2] := 0$). Returning to Eq. (9) we see:

$$\frac{\mathbb{E}\left[\left\langle \hat{y}_e, \hat{y}_o e^{-2\pi i \langle a, k/N\rangle}\right\rangle_r\right] - \gamma_k}{\sqrt{\mathbb{E}[\parallel \hat{y}_e\parallel_r^2]\mathbb{E}[\parallel \hat{y}_o e^{-2\pi i\langle a,k/N\rangle}\parallel_r^2]}} = \frac{\lambda^2[k] - \lambda^2[k+\tilde{N}/2] + \sigma^2[k] - \sigma^2[k+\tilde{N}/2] - 4\gamma_k}{\lambda^2[k] + \lambda^2[k+\tilde{N}/2] + \sigma^2[k] + \sigma^2[k+\tilde{N}/2]} = \frac{\lambda^2[k] + \sigma^2[k] - 4\gamma_k}{\lambda^2[k] + \sigma^2[k]}, \tag{15}$$

underestimates the FSC (Fig. 2c). Finally, if neither assumption is met, the SFSC fails to approximate the FSC and tends to give an overestimate (Fig. 2d). These results underpin the behavior of the SFSC, whether or not it should be applied, and motivate the improvements to the algorithm described in the following sections which circumvent the assumptions.

**Accounting for colored noise in the SFSC.** Microscopy images are often contaminated by colored Gaussian noise. Specifically, in cryo-EM, noise is often modeled by a covariance matrix that is diagonal in the Fourier domain, but with entries that vary[17]. In this case, Assumption 1 is violated. From Eq. (10), the ESFSC approximates the EFSC only in the case of white Gaussian noise and should not be expected to match otherwise. However, in the scenario where the noise is not white but its distribution can be estimated, we can first *whiten* the measurement prior to computing the SFSC. Suppose that the Fourier transform of the noise distribution is $\hat{\epsilon} \sim \mathcal{N}(0, D(\sigma^2))$. We define the Fourier transform of the whitened noisy measurement $\hat{y}_w$ as:

$$\hat{y}_w = W^{-1/2}\hat{y}, \tag{13}$$

where $W = D(\sigma^2)$. By construction, the noise in $\hat{y}_w$ is white:

$$\hat{y}_w = W^{-1/2}\hat{x} + W^{-1/2}\hat{\epsilon} = \hat{x}_w + \hat{\epsilon}_w, \tag{14}$$

where $\hat{x}_w = W^{-1/2}\hat{x}$ and $\hat{\epsilon}_w = W^{-1/2}\hat{\epsilon}$, and $\text{Cov}[\hat{\epsilon}_w] = \text{Cov}[W^{-1/2}\hat{\epsilon}] = W^{-1/2}WW^{-1/2} = I$. While this transform changes the signal, the SSNR of the new signal is the same as the original one. Noise whitening of data has statistical justifications which are described in ref. [18].

To demonstrate the effect of colored noise on the SFSC, we generated an image with a noise spectrum that decays following $\exp(-B\parallel k\parallel^2/4VN)$, with $B = 50$ Å$^2$. We show in Fig. 3c that after applying a whitening transform, we can recover the FSC from the SFSC. While this procedure can always be done if the noise level can be estimated, we note that it also leads to a simpler scheme for estimating the SSNR. Specifically, if the noise variance can be estimated, the ratio of the noise variance subtracted from the power spectrum to the noise variance is, in expectation, also

where $\gamma_k$ is a value we have chosen. The above equation equals the desired EFSC in Eq. (4) when we set $\gamma_k = \frac{1}{4}\sigma^2[k]$. Importantly, this procedure only works after a whitening transform of the original measurement since the variance of the noise is known. We show in Fig. 3d that upsampling a whitened measurement with a slow decaying power spectrum recovers the expected correlation curve. The effect of upsampling, and more generally frequency filtering[19] prior to computing the SFSC, is discussed in Supplementary Note 6.

**Estimating resolution from a single cryo-EM map.** For a conventional 3-D reconstruction pipeline in single particle cryo-EM, the data are split into random half sets to generate two independent half maps which are used to compute the FSC. To verify the assumptions and corrections introduced in this work, we show that the global resolution can be estimated from a single cryo-EM half map using the SFSC. We use the 3-D structures of a 20S proteasome (EMD-24822[20]), a 70S ribosome (EMD-13234[21]) and two small membrane proteins (EMD-27648[22], EMD-20278[23]) as examples.

After 3-D reconstruction, we do not expect the noise to be white, but instead to increase proportionally to the frequency due to the Fourier slice theorem, whereas the signal will show a strong exponential decay due to the B-factor. Thus, after whitening, our method applies. However, deposited maps usually have masking in either the volume or images which impact the noise statistics. Here we use the the noise in the corners of the reconstructions to estimate the noise distribution, but this assumes no masking was used. Otherwise, the noise in the corners and center will display different statistics. Specifically, we use the region outside a sphere which encompasses the molecular structure to estimate the noise by computing the spherically averaged power spectrum, defined as $\text{PS}(y)(r) := \int_{\mathcal{S}_r}|\hat{y}(\parallel k\parallel)|^2 dk$ (see Supplementary Fig. S4). We then apply the whitening transform and upsample procedure before computing the SFSC. The resolution reported using the standard FSC and the resolution calculated from the SFSC at a threshold of $1/7$[3] are approximately equal (Fig. 4a–c) except for the case where there is a non-unifom noise distribution (Fig. 4d). We attribute the deviation of the SFSC from one at the low

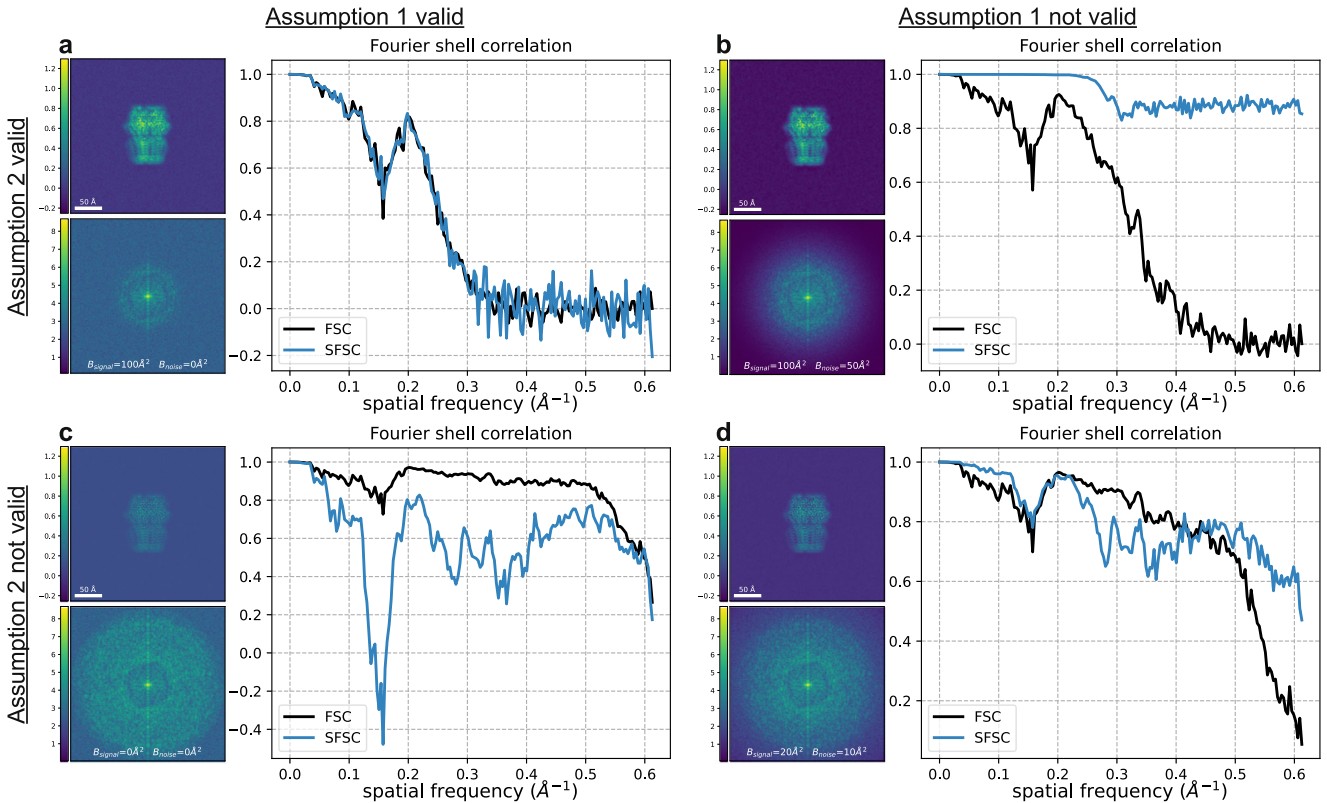

**Fig. 2 Conditions on the statistics of the signal and noise under which the SFSC accurately estimates the FSC.** Each panel shows the image, power spectrum and associated SFSC for a signal that satisfies or fails to satisfy both Assumption 1 and Assumption 2. The SNR was set to 15 for each image with additive Gaussian noise that decays with spatial frequency when specified. The FSC was computed for each case using two synthetic images generated with the same parameters but independent noise. **a** $B_{signal} = 100\,\text{Å}^2$, $B_{noise} = 0\,\text{Å}^2$. Both assumptions are met and the SFSC accurately estimates the FSC. **b** $B_{signal} = 100\,\text{Å}^2$, $B_{noise} = 50\,\text{Å}^2$. The noise is not white Gaussian and the SFSC overestimates the FSC. **c** $B_{signal} = 0\,\text{Å}^2$, $B_{noise} = 0\,\text{Å}^2$. The noise is white Gaussian but the signal does not have rapid decay. The SFSC underestimates the FSC. **d** $B_{signal} = 20\,\text{Å}^2$, $B_{noise} = 10\,\text{Å}^2$. Neither assumption is met and the SFSC fails to estimate the FSC. This figure demonstrates that the naive SFSC provides an accurate estimate of the FSC only if Assumption 1 and Assumption 2 are met.

frequencies to the difficulty in accurately estimating the noise using ad hoc methods. These results suggest that the SFSC provides a viable alternative for estimating resolution in cryo-EM and could be used in the absence of half maps.

**Denoising a reconstructed tomogram.** Having established the necessary assumptions and corrections under which the SFSC provides an estimate of the SSNR, we next demonstrate an application to denoising tomographic reconstructions from cryo-ET data. In a typical cryo-ET tilt series data collection scheme, projection images are measured at $\pm 60°$ in several degree increments. The recorded frames at each tilt are then motion corrected, aligned and a reconstruction technique is used to generate the 3-D tomogram. Due to the low electron beam dose required for imaging biological samples, the SSNR in cryo-ET data is low[24], and so there is a need for denoising methods[25]. Additionally, unlike in single particle cryo-EM, there are no related measurements to boost the SSNR by averaging. Thus, cryo-ET provides an ideal use case for the SFSC. Alternative approaches for estimating the resolution in cryo-ET such as computing the FSC from reconstructions of the even and odd images in a tilt series are described in[26].

Considering the model for a noisy measurement in Eq. (1), the minimum mean square error estimator for $x$ given $y$ under the Gaussian assumptions is known as the Wiener filter, and is widely used in cryo-EM[13,27,28]. The Wiener filter is defined as:

$$\hat{x}_{\text{WF}}(r) = \frac{1}{1 + \frac{1}{\text{SSNR}(r)}}\hat{y}(r). \qquad (16)$$

In cryo-ET, a common practice is to provide an ad hoc SSNR for Wiener filtering, or simply to use a low-pass filter. However, given the relationship between the EFSC and SSNR in Eq. (5), and that the SFSC can estimate the FSC, we show that the SFSC provides a simple, data-driven method for applying a Wiener filter. Combining Eq. (16) with Eq. (5), we get that $\hat{x}_{\text{WF}}(r) = \text{SFSC}(r) \cdot \hat{y}(r)$. Effectively, each shell in Fourier space for measurement $y$ is weighted according to the correlation profile from the SFSC. The idea of self-Wiener filtering has already been suggested in the context of signal processing and is described in ref. [29].

Here we show in Fig. 5 that applying a Wiener filter from the computed SFSC improves the visibility of a reconstructed tomogram. The data in this example is *C. elegans* tissue from EMD-4869[30]. In order to accurately estimate the SSNR, we know from Assumption 1 that the noise must be white Gaussian. To estimate the noise variance for a subsection of the tomogram, we select a slice above the region of interest. We then compute the SFSC and apply the Wiener filter in Eq. (16). The resulting denoised section of the tomogram shows enhanced contrast over the original and a low-pass filtered version. Specifically, the ribosomes and membrane edges stand out from the background. We additionally consider CTF effects in Supplementary Fig. S5 and

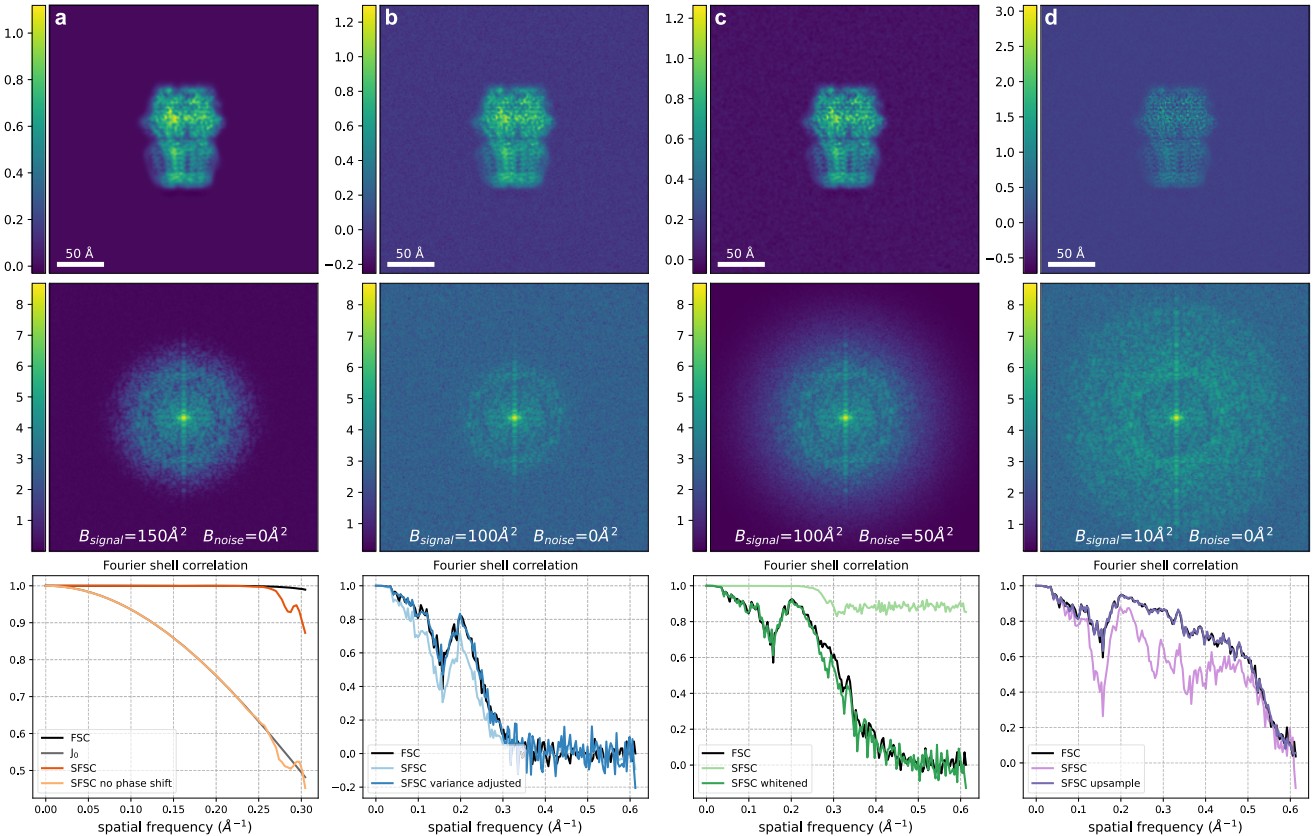

**Fig. 3 Corrections required for the SFSC to accurately estimate the FSC.** The image and corresponding power spectrum in each column were generated with a specified SNR and a B-factor on both the signal and noise to exemplify each case. The FSC was computed for each case using two synthetic images generated with the same parameters but independent noise. **a** Phase shift correction, $SNR = 10^5$, $B_{signal} = 150$ Å$^2$, $B_{noise} = 0$ Å$^2$. If the phase shift induced by downsampling is not corrected, the 2-D SFSC reduces to $J_0$, a scaled zeroth order Bessel function of the first kind (see Supplementary Note 4). **b** Correction for the scaled variance, $SNR = 15$, $B_{signal} = 100$ Å$^2$, $B_{noise} = 0$ Å$^2$. Both assumptions on the signal and noise are met. The SFSC estimates the FSC according to Eq. (12) after adjusting for the scaled variance. **c** Whitening transform, $SNR = 15$, $B_{signal} = 100$ Å$^2$, $B_{noise} = 50$ Å$^2$. After applying a whitening transform, the SFSC estimates the FSC. **d** Upsampling, $SNR = 10$, $B_{signal} = 10$ Å$^2$, $B_{noise} = 0$ Å$^2$. If the signal does not have rapid decay but has been whitened, the SFSC estimates the FSC only after upsampling. These correcting factors extend the applicability of the SFSC.

compare our approach to the noise learning method cryo-CARE[31] using a different data set (EMD-15056[32]) in Supplementary Fig. S6 and Supplementary Note 7. These results demonstrate that the SFSC can provide a simple, data-driven and parameter-free filter for improving the visualization of tomograms.

## Discussion

In this work, we derive a set of conditions and corrections required for accurately computing the Fourier shell correlation from downsampled versions of a single noisy measurement. We demonstrate that we are able to estimate the global resolution from a single map and denoise a reconstructed tomogram using the SFSC. Our approach is broadly applicable and allows for estimation of the SSNR if it is not possible to collect replicate measurements or use prior information. Furthermore, our approach does not require instrument specific calibration as described in[8]. The corrections we introduce in this work extend the applicability of the SFSC but also suggest a simpler path to estimating the SSNR provided an estimate of the noise can be obtained. We show that the same logic applies for any data processing pipeline in cryo-EM that estimates the noise or computes half maps (see Supplementary Note 5). If the noise cannot be accurately estimated or is non-uniform, then the SFSC should not be expected to work. While we estimate the noise with ad hoc methods here, using more accurate approaches typically

employed in cryo-EM data processing pipelines could improve the SFSC and associated Wiener filter.

Although the SFSC is not always applicable, there are many situations that can benefit from having an estimate of the SSNR from a single measurement. For example, in single particle cryo-EM, there are a growing number of methods which generate 3-D structures from manifold embeddings and do not produce independent half maps with which to compute the standard FSC[33–36]. Thus there is a need for alternative methods to estimate signal and noise statistics. In principle, the SFSC could also be used to circumvent splitting data into half sets during 3-D reconstruction. This has the potential to lead to improved reconstructions due to an increase in SSNR from using the full data set. One additional application of the SFSC to single particle cryo-EM could be to both denoise and estimate the resolution of 2-D class averages.

Other useful applications of the SFSC could include validation of SSNR enhancement after modification by neural network based approaches[37]. Similarly, the SFSC could provide an alternate measurement of the SSNR for noise learning based methods[31,38,39]. While the noise learning approaches give impressive results for denoising tomograms (see Supplementary Fig. S6), the benefit of the Wiener filter presented here is that it is fast to compute, requires no parameter tuning, and does not require extra storage (e.g., from reconstructing tomograms using odd and even frames). Further analysis of the SFSC for use with

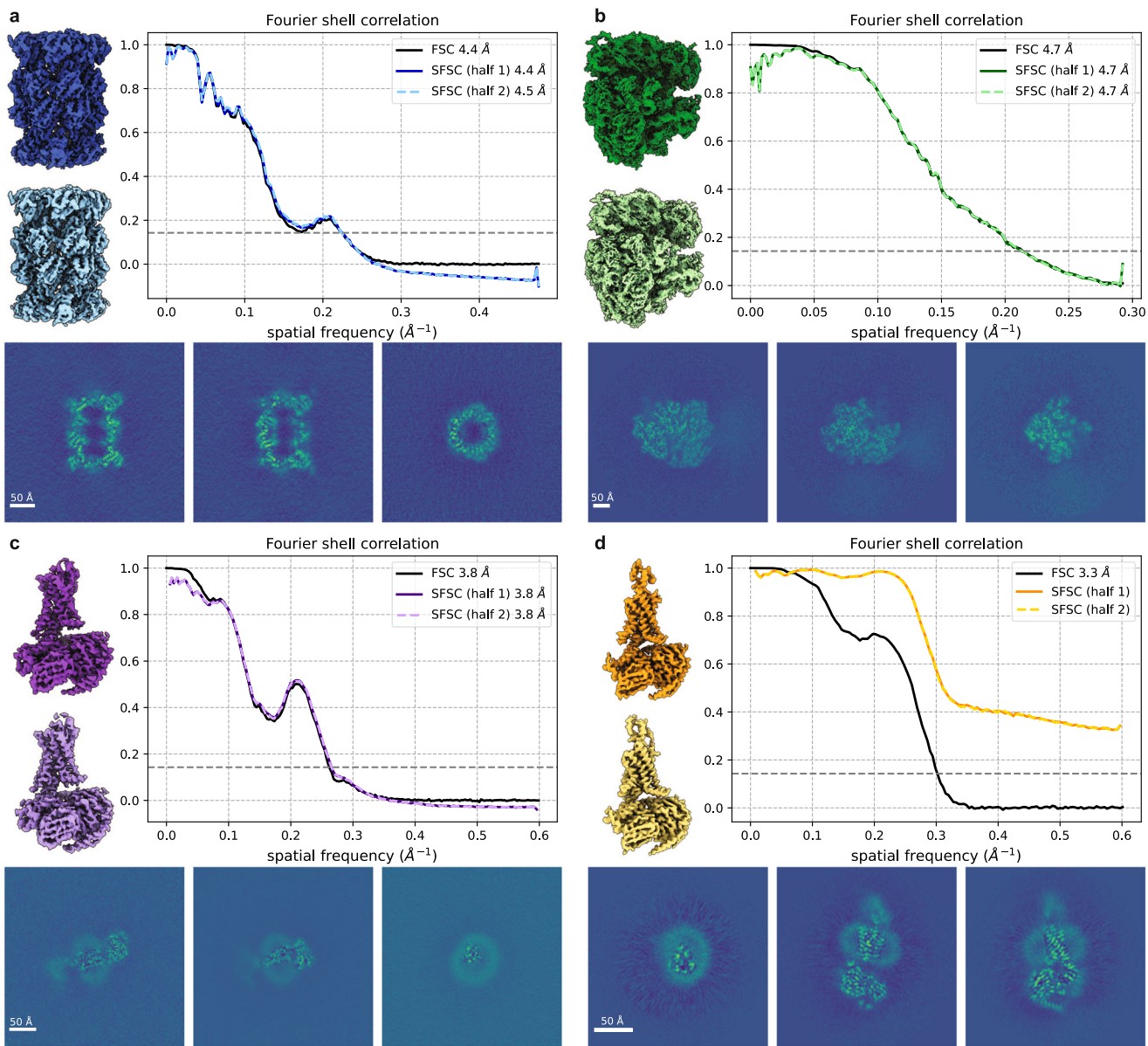

**Fig. 4 Global resolution estimates from single maps.** The SFSC is computed for each half map after applying the noise whitening and upsampling procedure. The noise is estimated by computing the spherically averaged power spectrum from the region outside a sphere encompassing the structure. The SFSC is approximately equal to the standard FSC for **a** EMD-24822 (grid points = $360^3$, voxel size = 1.05 Å), **b** EMD-13234 (grid points = $336^3$, voxel size = 1.7 Å) and **c** EMD-27648 (grid points = $416^3$, voxel size = 0.83 Å), but fails for **d** EMD-20278 (grid points = $288^3$, voxel size = 0.83 Å) due to the non-uniform noise which can be seen in the central slice images.

cryo-ET could also account for the missing wedge as well as directional and local resolution effects.

## Methods

**Main algorithm.** The algorithm presented in this work consists of 3 main steps and 3 preprocessing steps, depending on the properties of the measured signal. The main steps are parameter-free and can be written succinctly as: (1) for each dimension, split the measured signal into even and odd index terms along that dimension, (2) for each pair of downsampled measurements, compute the SFSC, and (3) average the SFSC from all pairs.

**Data preprocessing.** Prior to computing the main algorithm, the user should first discern if Assumption 1 and Assumption 2 are met. This can be checked by plotting the spherically averaged

power spectrum. If both assumptions are met, the power spectrum at the latter half of spatial frequencies should appear approximately constant. However, if the required assumptions are not met, the preprocessing steps described in this work should be applied. These steps can be applied regardless of the signal and noise properties as long as an estimate of the noise variance can be obtained. The preprocessing steps are: (1) estimate the the noise variance, (2) whiten the measured signal, (3) upsample the whitened signal.

**Estimating the noise variance.** Several strategies exist to estimate the noise variance from data. This estimate is required for computing the SFSC if the noise is not white Gaussian. For the case of estimating the noise variance from a half map of a 3-D reconstruction in cryo-EM, we use an ad hoc approach by taking the region outside a sphere encompassing the molecular structure. For example, with EMD-24822, we use a spherical mask with

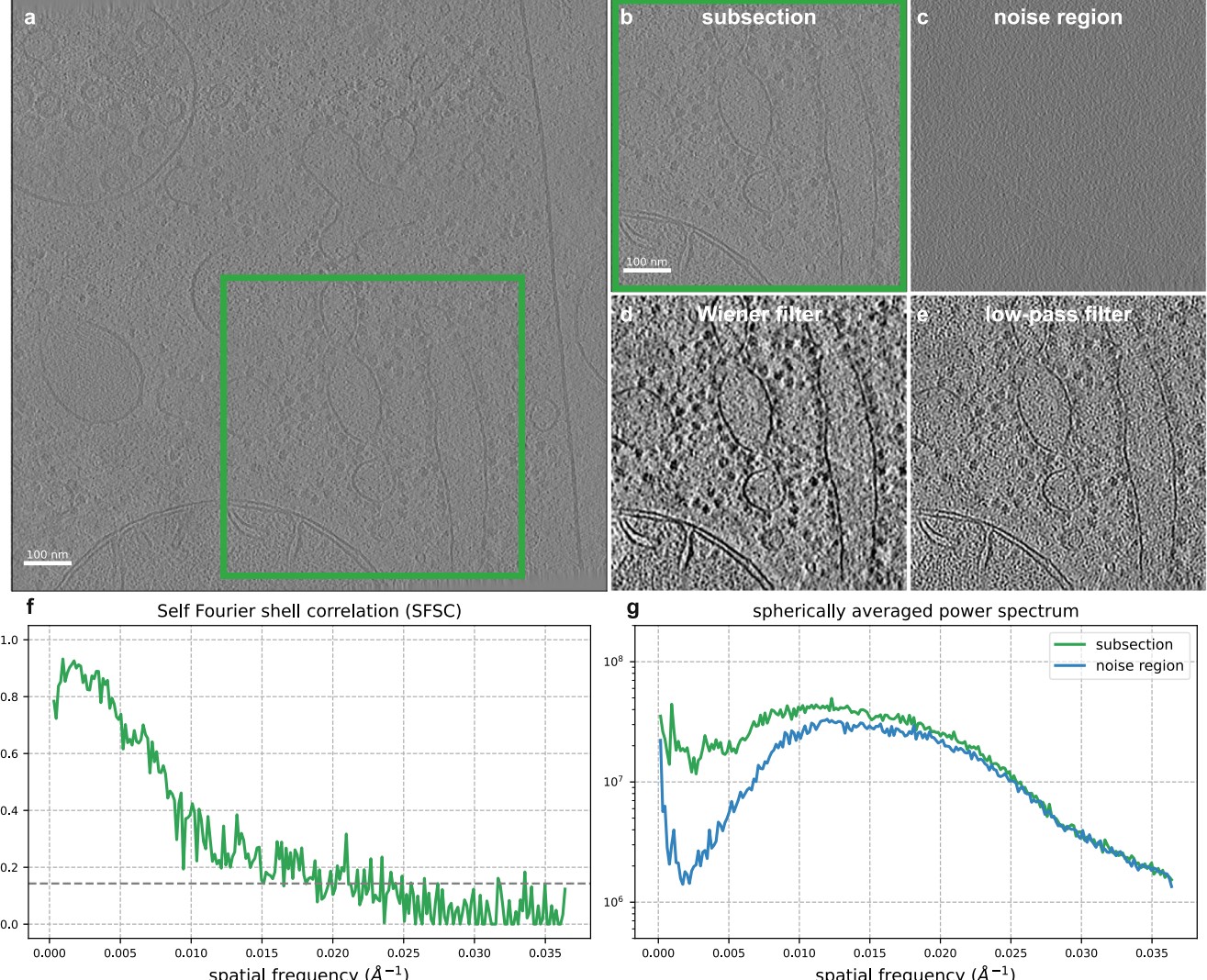

**Fig. 5 Denoising a reconstructed tomogram using the SFSC. a** Slice of a reconstructed tomogram of *C. elegans* tissue from EMD-4869 ($N \times N = 928 \times 928$, pixel size $= 13.7$ Å). **b** Region of interest from a subsection of the tomogram ($N \times N = 464 \times 464$). **c** Slice of the tomogram selected vertically above the region of interest containing background noise. **d** Slice from the region of interest after applying a Wiener filter. **e** Conventional low-pass filter of the subsection at 66 Å; determined using the 1/7 threshold of the SFSC. Both the Wiener filtered and low-pass filtered images are displayed at a threshold of ±2 standard deviations of the pixel values. **f** SFSC computed from the tomogram subsection. **g** Spherically averaged power spectrum of the region of interest slice and the background noise slice. The Wiener filter computed from the SFSC provides significantly increased contrast compared to a low-pass filtering approach.

radius $= 115$ Å. The noise variance is then estimated by computing the spherically averaged power spectrum of the noise region. However, we note that deposited maps usually have masking in either the volume or images which impact the noise statistics, and so this approach does not always apply.

**Denoising**. To estimate the noise variance for a reconstructed 3-D tomogram, we use a slice of the tomogram above the region of interest. Although the noise slice does not reflect the true 3-D noise profile, and more accurate methods can be used, we show that it is a suitable estimate for whitening based on the results of Wiener filtering. When applying the Wiener filter, we find that an additional low-pass filter at the spatial frequency corresponding to the 1/7 threshold from the SFSC can subtly enhance the contrast.

**Reporting summary**. Further information on research design is available in the Nature Portfolio Reporting Summary linked to this article.

## Data availability

All data sets used in this work are available from the Electron Microscopy Data Bank[40]. The entries used are EMD-11657, EMD-24822, EMD-13234, EMD-27648 and EMD-20278 for molecular structures, EMD-4869 and EMD-15056 for reconstructed tomograms, and EMPIAR-11058 for tilt series image.

## Code availability

The source code used to produce the results and figures in this work is available at github. com/EricVerbeke/self_fourier_shell_correlation, and is also deposited in Zenodo[41]. The SFSC will be made available as a tool in the software package ASPIRE[42].

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

## Acknowledgements

E.J.V., M.A.G. and A.S. are supported in part by AFOSR under Grant FA9550-20-1-0266, in part by Simons Foundation Math+X Investigator Award, in part by NSF under Grant DMS-2009753, and in part by NIH/NIGMS under Grant R01GM136780-01. T.B. is partially supported by the NSF-BSF award 2019752, the BSF grant no. 2020159, and the ISF grant no. 1924/21. We thank Zunlong Ke and Ricardo D. Righetto for valuable discussion and insight regarding cryo-ET.

## Author contributions

E.J.V. and A.S. conceived of this project. E.J.V. developed the software and ran numerical experiments. E.J.V., M.A.G., T.B. and A.S. designed the experiments and analyzed the results. E.J.V. and M.A.G. wrote the manuscript. All authors commented on and edited the manuscript.

## Competing interests

The authors declare no competing interests.
