## [Peer Review File · Communications Biology]

Reviewers' comments:

Reviewer #1 (Remarks to the Author):

Verbeke et al present an interesting idea, arguing that a Fourier shell correlation curve can be obtained with only one set of measurements of an object (e.g. 2D image or 3D reconstruction), instead of the conventional two sets of measurements. They argue for satisfying several conditions in order for the self FSC to be equivalent to the FSC between two independent measurements. This idea is really interesting, and I suspect will be important to the field in general. As the authors elaborate in the Discussion, although it may not replace the conventional FSC, it has a lot of potential for different types of applications. This work should be published. My comments are mostly intended as suggestions to express interest in the authors' idea, and to strengthen the conclusions for readers, especially for experimentalists.

For completeness, please show the SFSC from both half1 and half2 in Figure 4 (currently only half1 is shown).

Being an experimentalist, I would like the authors to provide a few more examples of the SFSC, and how it compares to the FSC of two independently derived half maps. I would like to suggest to repeat the procedure in Figure 4 to (1) a map of the ribosome and (2) something small, like a membrane protein, in addition to the 20S proteasome that they currently show. The ribosome is large and has lots of contrast; (2) a small membrane protein would be an example of a challenging protein assembly that is commonly encountered by experimentalists. Having two additional examples, of distinct types of protein assemblies, would give experimentalists more confidence that the idea of the SFSC could be applied more routinely.

I am not used to looking at tomography data, but I am curious as to why the lowpass filtered slice of the tomogram doesn't look so different from the simple reconstruction. I would expect it to look more contrasty. There are also other ways to alter contrast, including recent denoising algorithms (e.g. implemented in WARP), which would make the tomogram appear to have better contrast to the human eye. What I would like the authors to comment on is: how does the Wiener filtering approach implemented here compare with other denoising methods? How is the information content affected?

Reviewer #2 (Remarks to the Author):

In this manuscript by Verbeke et al., the authors come up with a new way to estimate FSC based resolution of 3D reconstructed maps (both single particle and tomography). They adopt this methodology from prior work by Koho et al., where they described an FRC based technique for image restoration in light microscopy. The advantage of the methodology described in this manuscript is that one can estimate FSC from a single 3D volume with the unique approach of splitting the volume and grouping the voxels. The authors have demonstrated the application of this methodology for single particle by using a 20S proteasome dataset (EMD-24822), GABA receptor dataset (EMD-11657) and also in tomography (by using the EMD-4869 dataset). While in the case of tomography this approach will be extremely beneficial when it comes to filtering of volumes and denoising, the effective utilization and advantage of this methodology in estimating FSC based resolution for single particle is not clear. A systematic benchmarking of the enhancement in speed of single particle 3D processing by using this technique versus splitting the data set into two half sets and carrying out simultaneous, independent 3D reconstructions would be useful to demonstrate the advantage of this methodology over existing techniques. The authors should address this in the revised manuscript. Otherwise this is

a new and unique methodology for estimating self FSC based resolution from single 3D reconstruction, and if properly implemented can be of great value to the field of structure determination by cryo-EM.

Reviewer #3 (Remarks to the Author):

The manuscript by Verbeke and colleagues evaluates the application of "self-FSC" measurements previously reported elsewhere for the resolution estimation of noisy volumes which typically are produced in the cryo-EM field. The authors introduce a novel strategy to downsample the signal of a volume which allows correlation of FSC between them. The authors note that there are two conditions that need to be satisfied in order for the measurement by SFSC to be meaningful, which are: 1) whiteness of the noise and 2) the fast decay of the radial power spectrum of the signal. The authors then test their SFCS measurement on synthetic data, an EMDb deposition and demonstrate high similarity between the FSC between the independent half-maps and the SFSC. Finally, the authors suggest the use of SFSC for filtering of electron cryotomograms.

The manuscript introduces an independent measurement of the resolution without the need for a statistically independent measurement. This is very useful as can be used for i.e. automated annotation or filtering of volumes. Overall, the manuscript definitively advances the toolset for the data analysis of cryo-EM volumes. I have three important comments which should be addressed before the manuscript could be published.

Major comments:

1. The method is designed and tested without the account for the CTF. Indeed, single particle structures are CTF corrected. However, CTF modulates the amplitude of the signal and may introduce correlation between the neighboring "Fourier Shells". Tomograms, even if CTF corrected, still have "zeros" of CTF which could introduce the correlation between the Fourier Shells. I suppose that in the case of the tested tomogram, the resolution was before the first zero of CTF, however this may not generally be the case. I recommend that the authors incorporate the considerations of CTF into their simulation and evaluate its effects.
2. The part of estimating the resolution from a simple map is based on just one structure. As the method seems not very demanding – the evaluation should be done more systematically on a large number of structures which are available in the structural databanks.
3. An unsolved discussion in the cryo-EM community that the later Fourier Shells contain more voxels and therefore the statistical significance threshold is higher at lower frequencies. When the resolution of a map is over half-Nyquist, the values for the threshold appear to be similar. However, in the case of SFSC the condition 2 suggests that many of the volumes should have the resolution before half-Nyquist. This consideration, for example, will result in a lower resolution estimate for a tomogram presented in Figure 5. Would the authors want to comment on the threshold for the resolution?

Minor points:

1. A label in Figure 5B says "subtomogram". This term is typically used as a part of "subtomogram averaging" and then it would need to contain a single molecule of interest for alignment and averaging. Consider revising.
2. It is not clear how the low-pass filter was performed, why 50 Å was chosen, not $1/0.015$ $1/\text{Å}$ where the FSC first touches the horizontal threshold in Figure 5F? It's obvious that the incorporation of higher frequencies will make the low-resolution features look noisier. Please provide a rationale for choosing the value for the low-pass filtering.

Response to Reviewers:

We thank the 3 referees and the editor for taking the time and effort to review our manuscript. The comments and suggestions were constructive and we think they have strengthened the manuscript overall. Our responses follow the remarks of each referee in blue text.

Reviewer #1 (Remarks to the Author):

Verbeke et al present an interesting idea, arguing that a Fourier shell correlation curve can be obtained with only one set of measurements of an object (e.g. 2D image or 3D reconstruction), instead of the conventional two sets of measurements. They argue for satisfying several conditions in order for the self FSC to be equivalent to the FSC between two independent measurements. This idea is really interesting, and I suspect will be important to the field in general. As the authors elaborate in the Discussion, although it may not replace the conventional FSC, it has a lot of potential for different types of applications. This work should be published. My comments are mostly intended as suggestions to express interest in the authors' idea, and to strengthen the conclusions for readers, especially for experimentalists.

We thank the reviewer for their assessment of our manuscript.

For completeness, please show the SFSC from both half1 and half2 in Figure 4 (currently only half1 is shown).

Figure 4 has now been updated to show the SFSC for both half maps.

Being an experimentalist, I would like the authors to provide a few more examples of the SFSC, and how it compares to the FSC of two independently derived half maps. I would like to suggest to repeat the procedure in Figure 4 to (1) a map of the ribosome and (2) something small, like a membrane protein, in addition to the 20S proteasome that they currently show. The ribosome is large and has lots of contrast; (2) a small membrane protein would be an example of a challenging protein assembly that is commonly encountered by experimentalists. Having two additional examples, of distinct types of protein assemblies, would give experimentalists more confidence that the idea of the SFSC could be applied more routinely.

We now include a comparison of the FSC and SFSC for three additional structures from the EMD. The new structures include a 70S ribosome (EMD-13234) and two different small membrane proteins (EMD-27648 and EMD-20278). Adding these additional structures highlights an important feature of the SFSC algorithm which is that in general, the SFSC should not be expected to work on deposited maps due to the potential non-uniform noise distribution after processing. Ideally, estimates of the noise computed during 3-D reconstruction would be used for the SFSC. We have added additional text to reflect the new Figure 4 and emphasize the point about uniform noise in Section 2.5 (pg. 5) and Section 3 (pg. 9).

I am not used to looking at tomography data, but I am curious as to why the lowpass filtered slice of the tomogram doesn't look so different from the simple reconstruction. I would expect it to look more contrasty. There are also other ways to alter contrast, including recent denoising algorithms (e.g. implemented in WARP), which would make the tomogram appear to have better contrast to the human eye. What I would like the authors to comment on is: how does the

Wiener filtering approach implemented here compare with other denoising methods? How is the information content affected?

To address the point about low-pass filtering, and also a comment by reviewer #3, we have updated the low-pass filtered image in Figure 5e to be filtered at the resolution when the SFSC reaches the 1/7 threshold (now low-pass filtered at 66 Å instead of 50 Å).

As to why the image does not appear to have more contrast, in general it is common to rescale images (automatically or manually) to improve visualization. In the original manuscript, we chose not to scale the values for comparative purposes. However, in the revised manuscript, we auto threshold each filtered image at ± 2 standard deviations of the pixel values. This significantly improves the contrast of the Wiener filter and also improves the low-pass filter. Rescaling of pixel values will ultimately be up to the user for preferred visualization. This update is now reflected in the text for Figure 5 (pg. 8).

We additionally include a new Figure A6 in the Appendix to compare the SFSC based Wiener filter to a tomogram denoised using cryo-CARE (EMD-15056). In this example, while the Wiener filter improves contrast for features like membrane edges and ribosomes, cryo-CARE excels at both suppressing background and enhancing relevant high-frequency information. Importantly, the Wiener filter is a statistically optimal filter and cannot hallucinate features, which is not true for neural networks. We have updated the text to reflect this in Section 3 (pg. 9).

Reviewer #2 (Remarks to the Author):

In this manuscript by Verbeke et al., the authors come up with a new way to estimate FSC based resolution of 3D reconstructed maps (both single particle and tomography). They adopt this methodology from prior work by Koho et al., where they described an FRC based technique for image restoration in light microscopy. The advantage of the methodology described in this manuscript is that one can estimate FSC from a single 3D volume with the unique approach of splitting the volume and grouping the voxels. The authors have demonstrated the application of this methodology for single particle by using a 20S proteasome dataset (EMD-24822), GABA receptor dataset (EMD-11657) and also in tomography (by using the EMD-4869 dataset). While in the case of tomography this approach will be extremely beneficial when it comes to filtering of volumes and denoising, the effective utilization and advantage of this methodology in estimating FSC based resolution for single particle is not clear.

We thank the reviewer for noting the potential benefit of our approach to tomography. We demonstrate the use of the SFSC for resolution estimation in single particle because it is sometimes useful, and so it is important to know when it works. Additionally, we highlight in the discussion that there are an increasing number of reconstruction methods (in particular for heterogeneity analysis) that do not generate half maps for the traditional FSC. We have modified the discussion to highlight our contribution in developing the theory of the SFSC generally, as well as suggesting other applications to single particle cryo-EM (e.g. 2-D class average resolution estimation and denoising). See updated text in Section 3 (pg. 9).

A systematic benchmarking of the enhancement in speed of single particle 3D processing by using this technique versus splitting the data set into two half sets and carrying out simultaneous, independent 3D reconstructions would be useful to demonstrate the advantage of

this methodology over existing techniques. The authors should address this in the revised manuscript.

The reviewer raises an interesting point - if one can estimate the FSC through the SFSC, then splitting the data into half sets may be unnecessary. While the computational runtime would be roughly the same, there is a potential benefit that the SSNR of a reconstruction would increase from using the full set of images instead of half sets. Unfortunately, implementing such a change in a program like RELION would be a large software undertaking. However, we have updated the discussion to incorporate the reviewer's suggestion for future investigation (pg. 9).

Otherwise this is a new and unique methodology for estimating self FSC based resolution from single 3D reconstruction, and if properly implemented can be of great value to the field of structure determination by cryo-EM.

Reviewer #3 (Remarks to the Author):

The manuscript by Verbeke and colleagues evaluates the application of "self-FSC" measurements previously reported elsewhere for the resolution estimation of noisy volumes which typically are produced in the cryo-EM field. The authors introduce a novel strategy to downsample the signal of a volume which allows correlation of FSC between them. The authors note that there are two conditions that need to be satisfied in order for the measurement by SFSC to be meaningful, which are: 1) whiteness of the noise and 2) the fast decay of the radial power spectrum of the signal. The authors then test their SFCS measurement on synthetic data, an EMDB deposition and demonstrate high similarity between the FSC between the independent half-maps and the SFSC. Finally, the authors suggest the use of SFSC for filtering of electron cryotomograms.

The manuscript introduces an independent measurement of the resolution without the need for a statistically independent measurement. This is very useful as can be used for i.e. automated annotation or filtering of volumes. Overall, the manuscript definitively advances the toolset for the data analysis of cryo-EM volumes. I have three important comments which should be addressed before the manuscript could be published.

We thank the reviewer for their comments and their analysis of our work.

Major comments:

1. The method is designed and tested without the account for the CTF. Indeed, single particle structures are CTF corrected. However, CTF modulates the amplitude of the signal and may introduce correlation between the neighboring "Fourier Shells". Tomograms, even if CTF corrected, still have "zeros" of CTF which could introduce the correlation between the Fourier Shells. I suppose that in the case of the tested tomogram, the resolution was before the first zero of CTF, however this may not generally be the case. I recommend that the authors incorporate the considerations of CTF into their simulation and evaluate its effects.

We agree with the reviewer that it is important to consider the effect of the CTF on the SFSC. Our manuscript now contains a new Appendix Section 1 which includes the CTF in the forward model. We demonstrate that given two signals modified by the same CTF (as is the case when

computing the SFSC), then the FSC can still be used to estimate the SSNR, but of the modified signal. Importantly, the SFSC still well approximates the FSC when there are CTF effects (see new Figure A1). However, estimating the resolution using the traditional 1/7 threshold for CTF modified signals may not be meaningful as the CTF causes oscillations of the FSC.

We provide additional visualization of CTF effects for tomography data in a new Appendix Figure 5. We show that Thon rings can be clearly visualized in tilt series images but are harder to detect in reconstructed tomograms. We note that even if the signal is modified by the CTF, the Wiener filter still provides a statistically optimal filter for denoising.

2. The part of estimating the resolution from a simple map is based on just one structure. As the method seems not very demanding – the evaluation should be done more systematically on a large number of structures which are available in the structural databanks.

We have now included three additional structures in Figure 4 (see also response #3 to reviewer #1). While we would like to test our method using structural databanks, the numerous data processing strategies used in cryo-EM prevent us from being able to do a systematic test. Under our assumptions, the unprocessed 3-D reconstructions could be evaluated using the SFSC. However, the deposited maps are often processed in different ways (e.g. masking is applied to either the images or volumes), which make it difficult to estimate the noise statistics.

3. An unsolved discussion in the cryo-EM community that the later Fourier Shells contain more voxels and therefore the statistical significance threshold is higher at lower frequencies. When the resolution of a map is over half-Nyquist, the values for the threshold appear to be similar. However, in the case of SFSC the condition 2 suggests that many of the volumes should have the resolution before half-Nyquist. This consideration, for example, will result in a lower resolution estimate for a tomogram presented in Figure 5. Would the authors want to comment on the threshold for the resolution?

This is an interesting point which we do not think we will be able to resolve here. The statistical significance of the FSC at low resolution is suspect, although, in practical scenarios, the SSNR is very high at low resolution, so the FSC score is not relevant. Regarding that the SFSC assumption 2 suggests resolution should be before half-Nyquist, we note that in Section 2.4 and Figure 3d we address how to compute the SFSC when the SSNR is large at high-frequencies. In particular the upsampling procedure we introduce allows for estimation of the resolution up to the Nyquist limit (same as with the standard FSC).

Minor points:

1. A label in Figure 5B says “subtomogram”. This term is typically used as a part of “subtomogram averaging” and then it would need to contain a single molecule of interest for alignment and averaging. Consider revising.

We thank the reviewer for the clarification and have updated “subtomogram” to “subsection” in the figures and text.

2. It is not clear how the low-pass filter was performed, why 50 Å was chosen, not 1/0.015 1/Å where the FSC first touches the horizontal threshold in Figure 5F? It's obvious that the incorporation of higher frequencies will make the low-resolution features look noisier. Please provide a rationale for choosing the value for the low-pass filtering.

We agree that the heuristic choice of 50 Å is arbitrary and have now updated the low-pass filter to the 1/7 threshold as suggested (see also response #4 to reviewer #1).

REVIEWERS' COMMENTS:

Reviewer #1 (Remarks to the Author):

The authors have addressed all the concerns that I brought up for the original submission. The updated manuscript is of significant interest and should be published.

Reviewer #2 (Remarks to the Author):

The authors have satisfactorily addressed all the concerns and comments by reviewers.

Reviewer #3 (Remarks to the Author):

The authors have significantly enhanced the revised manuscript.

I have a remaining concern regarding the resolution determination of single-particle cryo-EM maps. Since most EMDB entries in recent years include unfiltered, unmasked half-maps, I am unclear about the technical challenges associated with conducting FSC vs SFSC analysis on a larger dataset of structures.

Notably, the authors highlight an interesting outlier in Figure 4d, where structured noise amplifies SFSC. While I find the interpretation of the map containing structured noise compelling, establishing clear guidelines for reliably employing the SFSC analysis method is crucial for its widespread adoption by the scientific community. In my view, conducting FSC vs SFSC analysis on a substantial number of deposited structures and subsequently investigating which proteins or structure determination methods lead to outliers would be highly beneficial.